# Burnout and depression: A cross sectional study among health care workers providing HIV care during the COVID-19 pandemic in Malawi

**Khumbo Phiri**[1]*, **John Songo**[1], **Hannah Whitehead**[2], **Elijah Chikuse**[1], **Corrina Moucheraud**[3], **Kathryn Dovel**[1,2], **Sam Phiri**[1,4], **Risa M. Hoffman**[2], **Joep J. van Oosterhout**[1,2]

**1** Partners in Hope, Lilongwe, Malawi, **2** Department of Medicine, David Geffen School of Medicine, University of California, Los Angeles, Los Angeles, California, United States of America, **3** Department of Health Policy and Management, Fielding School of Public Health, University of California, Los Angeles, Los Angeles, California, United States of America, **4** Department of Public Health, School of Public Health and Family Medicine, Kamuzu University of Health Sciences, Lilongwe, Malawi

* khumbo@pihmalawi.com

**Data Availability Statement:** While we understand the journal's requirement of making all data related to this manuscript publicly available, we are

## Abstract

Health care workers (HCWs) in eastern Africa experience high levels of burnout and depression, and this may be exacerbated during the COVID-19 pandemic due to anxiety and increased work pressure. We assessed the prevalence of burnout, depression and associated factors among Malawian HCWs who provided HIV care during the COVID-19 pandemic. From April-May 2021, between the second and third COVID-19 waves in Malawi, we randomly selected HCWs from 32 purposively selected PEPFAR/USAID-supported health facilities for a cross-sectional survey. We screened for depression using the World Health Organization Self Report Questionnaire (positive screen: score≥8) and for burnout using the Maslach Burnout Inventory tool, (positive screen: moderate-high Emotional Exhaustion and/or moderate-high Depersonalization, and/or low-moderate Personal Accomplishment scores). Logistic regression models were used to evaluate factors associated with depression and burnout. We enrolled 435 HCWs, median age 32 years (IQR 28–38), 54% male, 34% were clinical cadres and 66% lay cadres. Of those surveyed, 28% screened positive for depression, 29% for burnout and 13% for both. In analyses that controlled for age, district, and residence (rural/urban), we found that screening positive for depression was associated with expecting to be infected with COVID-19 in the next 12 months (aOR 2.7, 95%CI 1.3–5.5), and previously having a COVID-19 infection (aOR 2.58, 95CI 1.4–5.0). Screening positive for burnout was associated with being in the clinical cadre (aOR 1.86; 95% CI: 1.2–3.0) and having a positive depression screen (aOR 3.2; 95% CI: 1.9–5.4). Reports of symptoms consistent with burnout and depression were common among Malawian HCWs providing HIV care but prevalence was not higher than in surveys before the COVID-19 pandemic. Regular screening for burnout and depression should be encouraged, given the potential for adverse HCW health outcomes and reduced work performance. Feasible interventions for burnout and depression among HCWs in our setting need to be introduced urgently.

unfortunately unable to comply with this requirement due to the reasons highlighted below. 1. Data contain potentially identifying or sensitive participant information. Even though the data for this manuscript are de-identified, the information that the study was conducted at Partners in Hope's supported sites may draw attention to individual health care workers at the health facilities. 2. Who has imposed the restriction of publicly sharing the manuscript data? The Malawi's National Health Sciences Research Committee approved our protocol in which under the ethical considerations section on protection of human subjects' privacy and confidentiality, we stated that participant information will not be released without written permission from the researchers, except as necessary for review, monitoring, and/or auditing. 3. Provide non-author contact information for a data access committee, ethics committee, or other institutional body to which data requests may be sent. Mr. Mackenzie Chivwara can be contacted through his email address (mackenzie@pihmalawi. com) for any data requests and Mr. Chivwara will then initiate a communication with the National Health Sciences Research Committee to ask for permission to release the de-identified data.

**Funding:** This work was supported by the United States Agency for International Development under Cooperative Agreement 72061221CA00010. All authors received salary support from this grant. The views in this publication do not necessarily reflect the views of the U. S. Agency for International Development (USAID), the U. S. President's Emergency Plan for AIDS Relief (PEPFAR) or the United States Government. The funders had no role in study design, data collection and analysis, decision to publish, or preparation of the manuscript.

**Competing interests:** The authors have declared that no competing interests exist.

## Introduction

Health systems in Africa require a motivated and effective health workforce but this is challenged by a high prevalence of mental health conditions among health care workers (HCWs) [1, 2]. Depression and burnout in particular threaten the wellbeing and effectiveness of HCWs [3, 4]. Evidence from before the COVID-19 pandemic shows high rates of psychological distress, anxiety, depression and burnout among HCWs in resource limited settings [4–7]. A systematic review from countries in Asia, South America, Europe and Africa, documented an overall prevalence of depression and depressive symptoms of 29% (range 21% to 43%) in the medical workforce [8]. Depression and burnout increase staff turnover, decrease quality of care provided to clients, and cause poor health outcomes among HCWs [9–11]. Although depression and burnout are distinct conditions, they have been shown to be related [12]. In a study conducted with nurses in South Africa, the presence of burnout syndrome was a determinant of depression [13]. Burnout results from long-term job-related stress. Based on the Job-Demand Resource model [14], and evidence from other studies [15], it is hypothesized that experiencing burnout may lead to ill health, including depression. Research from before the COVID-19 pandemic found that burnout prevalence among HCWs in the sub-Saharan Africa region ranged from 40 to 80%, depending on country, type of HCW and the burnout measurement tool [16]. Workload may contribute to burnout [4]. In Malawi, the HIV prevalence among adults is 8.9%, and around 1,000,000 Malawians are living with HIV [17]. Of those, 88.3% are aware of their HIV-positive status and 97.9% are on antiretroviral therapy (ART) [17] This has created a huge care burden for HIV treatment clinics, which at the same time experience severe staff shortages, as only 28 nurses and two physicians are available per 100,000 population [18]. In such high-burden circumstances, HIV service providers may be at particularly high risk among HCWs to experience mental health problems. A Malawian study reported 62% burnout among HCWs in HIV care in 2018 [10].

Malawi registered its first confirmed COVID-19 case in April 2020. Several nation-wide measures were introduced, including school closure, COVID-19 screening at border posts and restricting attendance of public events [19]. The Malawi Ministry of Health provided a package of interventions at health facilities, such as increased hand washing, physical distancing, using facemasks and temporary suspension of some HIV services [19]. In March 2021, COVID-19 vaccination rollout started for priority populations. By May 2021, during the study period, around 35,000 confirmed cases and 1,200 deaths had been recorded, with an estimated case-fatality ratio of 3.4% [20]. For a variety of reasons, the COVID-19 pandemic may exacerbate mental health concerns among HCWs, including post-traumatic stress disorder, anxiety, depression and burnout [21]. Despite preventive measures, HCWs may worry about their own health and their elevated risk of contracting COVID-19 [22]. They may also have concerns about being a source of exposure for their family members [11]. Finally, the COVID-19 disease burden may have added pressures to already overstretched health systems [23].

At the time when the COVID-19 vaccine had just been rolled out in Malawi to HCWs as a priority population, we sought to assess the prevalence of depression and burnout and associated factors among HCWs working in HIV services during the COVID-19 pandemic in Malawi.

## Methods

### Study design, site and participant selection

We conducted a cross-sectional survey of 435 HCWs from 32 health facilities supported by Partners in Hope, a USAID/PEPFAR implementing partner that supports HIV/AIDS care and

treatment in Malawi. Selected sites included 25 public health facilities (6 urban and peri-urban district hospitals; 3 rural community hospitals; and 16 rural health centres) and 7 faith-based hospitals (3 urban and 4 rural) run by the Christian Health Association of Malawi. The health facilities were purposively selected across the three geographical regions of Malawi. We included facilities with a high HIV burden where COVID-19 greatly affected HIV service delivery.

The sample size was determined using a standard sample size formula for estimating proportions, where we included the prevalence of burnout derived from a study that was conducted in Malawi in 2018 [10]. Due to lack of data on the number of HCWs within different cadres at each of the 32 purposively selected facilities, which would be required to draw a sampling frame, we were not able to distribute this sample across the facilities proportionate to size, in order to have a self-weighted sample. Instead, the calculated sample size of 435 was equally distributed over the 32 health facilities, resulting in 14 HCW required per facility.

From April-May 2021, between the second and third COVID-19 waves in Malawi, we invited up to 14 HCWs per facility to participate in an anonymous survey. Eligibility criteria were: ≥18 years of age and providing outpatient HIV care during the three months preceding data collection. We stratified HCWs into clinical and lay cadres, then randomly selected one individual per stratum and repeated this until up to 14 health care workers were invited for participation per facility. Clinical cadres included physicians, clinical officers, medical assistants and nurses. Lay cadres were staff who support HIV counseling and treatment, health promotion, and community-based care (HIV Diagnostic Assistants, Health Surveillance Assistants, Treatment Supporters, and Data Clerks).

## Data collection tools

We used the World Health Organization's (WHO) Self Reporting Questionnaire (SRQ) [24] and the Maslach Burnout Inventory (MBI) [25], which are validated screening tools for depression and burnout, respectively [26]. We also collected socio-demographic information and asked questions about participants' personal experiences with COVID-19. We used a version of the MBI tool that was previously modified for use in Malawi, to ensure relevance and appropriateness of questions in the local setting [10]. The tool was translated into the local language (Chichewa) and administered by research assistants in private spaces. Responses were directly logged into SurveyCTO, a mobile data collection platform, using Android tablets.

All respondents provided oral consent to participate in the survey. We did not collect any personal identifying data. HCWs received their screening results and those who screened positive for depression and burnout were offered counseling by lay cadre staff and psycho-social counselors. Research assistants received guidance on how to refer respondents who indicated having severe depression symptoms to a mental health service provider. The study was approved by the Malawi National Health Sciences Research Committee (reference number 19/08/2338), and the University of California Los Angeles Institutional Review Board (reference number 21–000392).

## Measurement of outcomes

**Depression measurement.** The WHO SRQ tool is comprised of 20 yes/no questions about physical, mental and emotional symptoms in the past 30 days. In order to capture experiences during the last COVID-19 wave, we extended this period to 3 months. "Yes" responses were assigned 1 point, and "no" responses 0 points; the responses were summed (possible range: 0–20), and the threshold for a positive screen for depression was set at ≥8, which was validated in previous studies in Malawi [27, 28].

**Burnout measurement.** The validated MBI tool has 22 questions across three domains of burnout: emotional exhaustion (EE), which measures exhaustion at work; depersonalization (DP), which measures negative or cynical feelings; and low personal accomplishment (PA), which measures feelings about one's competencies. Each item asks how often respondents feel that way about their job, rated on a seven-point Likert scale (0–6). Sums were calculated for each of the three domains and scores were classified as low, moderate or high. Consistent with previous research conducted in Malawi [16], we defined a positive screen for burnout domains as having a moderate (19–26) or high (27+) score on EE, and/or a moderate (6–9) or high (10+) score on DP, and/or a moderate (39–34) or low (0–33) score on PA. Higher EE and DP scores, and a lower PA score correspond with greater burnout. Many of the items in the MBI ask specifically about feelings related to patient interactions. Because data clerks were not directly involved in healthcare provision, we excluded them from all analyses related to burnout.

## Data analysis

Our analysis had two main outcomes: depression and burnout. We used descriptive statistics to describe the prevalence of screening positive for depression and of screening positive for burnout. Burnout prevalence was determined on the basis of its individual constructs of EE, DP and PA. Multivariable logistic regression models were used to assess factors associated with positive screens for depression and burnout. Lastly, we used multivariable logistic regression to investigate the association between burnout and depression.

Regression models were fitted using a causal modelling approach [29] to investigate causal relationships between independent variables, selected based on assumptions drawn from literature, and the outcome variables. Independent variables of interest included COVID-19 experiences and risk perceptions, HCW cadres, marital status, and years of work [30, 31]. Multivariable logistic regression models controlled for age, district and residence (rural/urban) as potential confounders. All multivariable logistic regressions models used cluster robust standard errors that allowed for intragroup correlation at the facility level to control for clustering [32]. All analyses were conducted using Stata v17.

## Results

Of the 435 HCWs included, 54% were male and the median age was 32 (IQR: 28–38) years. Two-thirds were from rural health facilities. One-third were clinical cadres and the rest were lay cadres, of which 26% were data clerks. Thirteen percent reported ever having confirmed or suspected COVID-19, 15% had a household member, family member, or close friend who died from COVID-19-related complications, and 8% believed they would definitely or probably get COVID-19 in the next 12 months (Table 1).

### Prevalence and determinants of positive depression screen

Twenty-eight percent of respondents screened positive for depression, with a median SRQ score of 5.6 (IQR 3–8). Factors significantly associated with a positive depression score were: believing he/she will probably or definitely get COVID-19 in the next 12 months, and ever had suspected or confirmed COVID-19. Female HCWs had higher prevalence of positive depression screen than males (32% versus 25%), but the difference was not significant. Lay cadres, who constituted the majority (64%) of the HCWs who screened positive for depression, had a non-significantly lower prevalence of positive depression screen vs. clinical cadres (27% vs. 30%) (Table 2).

**Table 1. Description of HCW sample (n = 435).**

| Characteristic | N (%) or median (IQR) |
|---|---|
| Region | |
| Central | 220 (51%) |
| Southern | 176 (40%) |
| Northern | 39 (9%) |
| Location | |
| Rural | 299 (69%) |
| Urban | 136 (31%) |
| Cadre | |
| Non-clinical lay cadres | |
| - Data clerk | 75 (17%) |
| - Other* | 211 (49%) |
| Clinical cadres | 149 (34%) |
| Years worked in cadre | |
| Median (IQR) | 4 (1–8) |
| Sex | |
| Male | 236 (54%) |
| Female | 199 (46%) |
| Age | |
| Median, years (IQR) | 32 (28–38) |
| Marital Status | |
| Single | 114 (26%) |
| Married | 294 (68%) |
| Divorced/Separated/Widowed | 27 (6%) |
| *COVID-19 experiences and risk perceptions* | |
| Ever tested CoV-SARS-2 positive or had suspected COVID-19 | |
| No | 379 (87%) |
| Yes | 56 (13%) |
| Household member, family member, or close friend had COVID-19 | |
| No | 250 (58%) |
| Yes | 185 (42%) |
| Household member, family member, or close friend died of COVID-19 | |
| No | 370 (85%) |
| Yes | 65 (15%) |
| COVID-19 Risk Perception | |
| Believe will unlikely, probably not or definitely not have COVID-19 in next 12 months | 384 (92%) |
| Believe will definitely or probably have COVID-19 in next 12 months | 34 (8%) |

IQR: inter-quartile range.

* (Patient supporters, HIV diagnostic assistants and health surveillance assistants)

## Prevalence of burnout and factors associated with burnout

After excluding data clerks, 360 HCWs remained available for the burnout analyses. Of these, 29% screened positive for burnout. The prevalence was significantly higher among clinical cadres than lay cadres (37% vs. 23%, p = 0.01). No other factors were significantly associated with a positive burnout screen (Table 3).

**Table 2. Prevalence and determinants of positive depression screen.**

| Characteristic | Prevalence % (n) | ORs | (95% CI) | P-Value | aORs | (95% CI) | P-Value |
|---|---|---|---|---|---|---|---|
| Overall prevalence | **28.1 (122)** | | | | | | |
| Cadre | | | | | | | |
| Lay cadre* | 27.3 (78) | 1 | | | 1 | | |
| Clinical | 29.5 (44) | 1.12 | (0.7–1.7) | 0.62 | 1.15 | (0.7–1.8) | 0.55 |
| Years in cadre | | | | | | | |
| 0–5 year | 28 (75) | 1 | | | 1 | | |
| 6 or more years | 28 (47) | 0.99 | (0.6–1.5) | 0.98 | 1.03 | (0.6–1.7) | 0.90 |
| Gender | | | | | | | |
| Male | 25 (59) | 1 | | | 1 | | |
| Female | 31.7 (63) | 1.39 | (0.9–2.1) | 0.12 | 1.40 | (0.9–2.1) | 0.12 |
| Marital Status | | | | | | | |
| Single | 29.8 (34) | 1 | | | 1 | | |
| Married | 26.5 (78) | 0.85 | (0.5–1.4) | 0.50 | 0.89 | (0.5–1.5) | 0.66 |
| Divorced/Separated/Widowed | 37 (10) | 1.38 | (0.6–3.3) | 0.47 | 1.54 | (0.6–4.2) | 0.40 |
| Ever tested CoV-SARS-2 positive or had suspected COVID-19 | | | | | | | |
| No | 25.3 (96) | 1 | | | 1 | | |
| Yes | 46.4 (26) | 2.56 | (1.4–5.0) | <0.01 | 2.58 | (1.4–5.0) | <0.01 |
| Household member, family member, or close friend died of COVID-19 | | | | | | | |
| No | 102 (27.13) | 1 | | | 1 | | |
| Yes | 20 (33.9) | 1.38 | (0.8–2.5) | 0.28 | 1.42 | (0.8–3.0) | 0.25 |
| COVID-19 risk perception | | | | | | | |
| Believe will unlikely, probably not or definitely not have COVID-19 in next 12 months | 26.3 (101) | 1 | | | 1 | | |
| Believe will definitely or probably have COVID-19 in next 12 months | 47.1 (16) | 2.49 | (1.2–5.1) | 0.01 | 2.67 | (1.3–5.5) | 0.01 |

OR, odds ratio, aOR, adjusted odds ratio; CI, confidence interval.

*Patient Supporter, HIV Diagnostic Assistant, Health Surveillance Assistant, Data Clerk, *adjusting for Age, district and Urban/rural

## Prevalence of burnout by domains

Thirty-nine HCWs registered positive burnout scores in emotional exhaustion (3% high and 8% moderate). Commonly reported symptoms within this domain included feeling "used up" at the end of the workday and "working too hard on one's job". Only 1% of HCWs registered a high and 2% a moderate score in the depersonalization domain, with common symptoms reported as "feelings of becoming more unsympathetic towards people/clients" and "blaming clients for their problems". The highest prevalence (21%, n = 76) was in the personal accomplishment domain (4% low and 17% moderate), mostly from "feeling less energetic" and "not being able to deal with emotional problems calmly" (Table 4).

## Association between burnout and depression

Among the 104 client-facing HCWs who screened positive for burnout, nearly half also screened positive for depression (46%, n = 48). Controlling age, sex, marital status, years of work and geographical location, having a positive burnout screen was significantly associated with screening positive for depression (aOR: 3.2, 95%CI 1.9–5.4).

## Discussion

A year into the COVID-19 pandemic, after two waves of COVID-19 in Malawi and prior to the availability of widespread vaccination, we found that 28% of Malawian HCWs in HIV care

**Table 3. Prevalence and determinants of positive burnout screen.**

| Characteristic | Prevalence %(n) | ORs | (95% CI) | P-Value | aORs | (95% CI) | P-Value |
|---|---|---|---|---|---|---|---|
| Overall prevalence | **28.1 (122)** | | | | | | |
| Cadre | | | | | | | |
| Lay cadre* | 23 (49) | 1 | | | 1 | | |
| Clinical | 37 (55) | 1.92 | (1.2–3.1) | 0.01 | 1.86 | (1.2–3.0) | 0.01 |
| Years in cadre | | | | | | | |
| 0–5 year | 28 (55) | 1 | | | 1 | | |
| 6 or more years | 31 (49) | 1.17 | (0.7–1.9) | 0.49 | 1.03 | (0.6–1.8) | 0.91 |
| Sex | | | | | | | |
| Male | 26 (49) | 1 | | | 1 | | |
| Female | 32 (55) | 1.30 | (0.8–2.1) | 0.26 | 1.26 | (0.8–2.0) | 0.33 |
| Marital Status | | | | | | | |
| Single | 30 (34) | 1 | | | 1 | | |
| Married | 27 (78) | 0.98 | (0.6–1.7) | 0.93 | 0.86 | (0.5–1.6) | 0.63 |
| Divorced/Separated/Widowed | 37 (10) | 1.49 | (0.6–3.8) | 0.41 | 1.17 | (0.4–3.5) | 0.78 |
| Ever tested CoV-SARS-2 positive or had suspected COVID-19 | | | | | | | |
| No | 29 (92) | 1 | | | 1 | | |
| Yes | 28 (12) | 0.94 | (0.5–1.9) | 0.87 | 0.92 | (0.4–1.9) | 0.82 |
| Household member, family member, or close friend died of COVID | | | | | | | |
| No | 87 (28) | 1 | | | 1 | | |
| Yes | 17 (33) | 1.27 | (0.7–2.3) | 0.5 | 1.17 | (0.6–2.2) | 0.6 |
| COVID-19 risk perception | | | | | | | |
| Believe will unlikely, probably not or definitely not have COVID-19 in next 12 months | 28 (88) | 1 | | | 1 | | |
| Believe will definitely or probably have COVID-19 in next 12 months | 42 (11) | 1.93 | (0.9–4.3) | 0.11 | 1.89 | (0.8–4.3) | 0.13 |

OR, odds ratio, aOR, adjusted odds ratio; CI, confidence interval.

*Patient Supporter, HIV Diagnostic Assistant, Health Surveillance Assistant, Data Clerk *adjusting for Age, district and Urban/rural

screened positive for depression, 29% for burnout and 13% for both. A positive screen for depression was associated with reporting a previous COVID-19 infection, and with believing that one would get COVID-19 in the next 12 months. Screening positive for burnout was associated with being a clinical health worker (versus being a lay cadre staff). There was a strong association between screening positive for depression and burnout. The prevalence of depression we observed was lower compared to recent literature. A systematic review of studies among HCWs providing frontline COVID-19 care in 17 countries from Asia and Africa found

**Table 4. Prevalence of burnout by domain.**

| | Emotional Exhaustion (EE) | | | Depersonalization (DP) | | | Personal Accomplishment (PA)* | | |
|---|---|---|---|---|---|---|---|---|---|
| | Low | Moderate | High | Low | Moderate | High | High | Moderate | Low |
| BO score | 0–18 | 19–26 | 27+ | 0–5 | 6–9 | 10+ | 40+ | 39–34 | 0–33 |
| (n = 360) ^ | n = 321 | n = 30 | n = 9 | n = 350 | n = 7 | n = 3 | n = 283 | n = 62 | n = 14 |
| % | 89% | 8% | 3% | 97% | 2% | 1% | 79% | 17% | 4% |
| (95% CI) | (86–92%) | (6–12%) | (1–5%) | (95–99%) | (1–4%) | (0–3%) | (74–83%) | (14–22%) | (2–7%) |

^ data missing of one person

*For EE and DP, higher score corresponds with greater burnout. For the PA sub-scale, lower score corresponds with greater burnout. The grey-filled boxes correspond with positive burnout screen

the mean prevalence for depression was 43% [33]. HCWs included in our study did not provide frontline COVID-19 care. However, they were providing HIV care in clinics where they were likely exposed as they provided services to individuals who might have come in with symptoms of COVID-19. At the time of our study, a COVID-19 wave had just abated and vaccine roll out was on its way, with high uptake among HCWs by the end of May 2021 [34]. This may have created a more positive outlook for HCWs and may have mitigated COVID-19's role on the development of depressive symptoms and burnout.

Our study took place when there was uncertainty about the future of the pandemic, disruptions to personal networks had occurred [35, 36], and HCWs had experienced illness and bereavement among close family members and peers [37]. The Malawi government had imposed restrictions to control the spread of COVID-19 [20] and these limited interactions with people who could provide social support, which may cause negative psychological effects [38]. We found that previous COVID-19 infection and having a higher perceived risk of contracting COVID-19 were associated with positive depression screen. Previous studies also found that fear of COVID-19, either through personal experience or perceived future risk, impacts HCWs' mental wellbeing and can be a precursor of several mental disorders including anxiety and post-traumatic stress [30, 33, 37]. A study from Ethiopia documented that many HCWs were worried about contracting COVID-19, and found that this was positively associated with depressive symptoms and anxiety [39].

We found a relatively low prevalence of a positive screen for burnout (29%) compared to the broader literature from sub-Saharan Africa prior to the COVID-19 pandemic, when burnout rates among HCWs ranged between 40–80% and positivity was higher in all three domains [3]. Burnout in our population was largely driven by positivity in the personal accomplishment domain, indicating that HCWs had a low sense of competence and achievement in their work. In studies from Malawi more than a decade before the COVID-19 pandemic, the prevalence of burnout in the personal accomplishment domain was reported to be as high as 40% [16, 40]. In a 2018 study among HCWs providing HIV care in Malawi, using the same screening tool and criteria for burnout, the prevalence was 62% [10]. This study utilized self-administered written surveys, while ours used research assistants conducting in-person interviews with HCWs. This may have led to social desirability bias within our study, which could have contributed to a lower prevalence of burnout.

An important difference between our and previous studies is that two-thirds of our participants were non-clinical lay cadres. Compared to clinical cadres, lay health workers had a similar prevalence of positive screen for depression (30% vs. 27%) but a much lower prevalence of positive burnout screen (37% vs. 23%). Malawi's health care system has critical staff shortages and rising numbers of patients, increasing the burden of responsibility mostly among clinical health workers [18], which was underlined by the association of clinical cadre with burnout in our study. The majority of lay health workers among the participants in our study (64%) contributes to the lower observed overall burnout prevalence compared to previous studies.

We found that HCWs who screened positive for burnout were three times more likely to screen positive for depression. A study among nurses which examined the relationship between burnout and depression also found that burnout was related to increased prevalence of depressive symptoms [41]. While debate about distinction and overlap between burnout and depression continues [42], burnout is considered to be mostly work-specific and there is some evidence that burnout is a risk factor for depression [43–45]. Given these points, and because their management is different, it is important to screen HCWs for both conditions and to develop interventions that address the specific aspects of these conditions.

Evidence-based treatment for depression, which is feasible in low-resource settings, is available in Malawi, including pharmacotherapy and psychological interventions [46–48]. A variety

of counseling and psychological support interventions for burnout has been studied in Africa and some were found to be effective [49–51]. This evidence base still appears to be limited and more high-quality studies are needed for the management and prevention of burnout in our setting. If interventions for depression and burnout are to have meaningful impact, a large diagnostic and treatment gap for mental disorders, caused by scarce human resources, competing public health priorities, and low investment in mental health services, must be addressed [52].

Strengths of our study are the large sample, which was geographically representative and included clinical staff and lay cadres from rural and urban, and government and private health facilities. We also note several limitations. We used surveys that relied on a research assistant to ask each question and record the answer. As mentioned above, this may have caused measurement error due to social desirability bias, particularly for the MBI burnout scale, which is typically self-administered, and may have resulted in an underestimation of burnout. Our cross-sectional survey also cannot determine causality of the associations found. Results may have differed during COVID-19 waves, and we did not capture cumulative, longer-term COVID-19 effects. Due to lack of a priori data on the total number of HCWs within clinical and lay cadres at the selected 32 facilities that were needed to draw a sampling frame, we were unable to estimate a self-weighted sample that would consider multi-stage sampling. Alternatively, we used standard statistical inferencing assumptions, hence the generalization of our findings to the population level must be considered with caution.

## Conclusion

While HCWs who provide HIV care in Malawi screened positive for burnout and depression commonly, prevalence rates were not higher than reported before the COVID-19 pandemic. Our results reinforce the importance of screening HCWs for burnout and depression given the negative consequences for mental health and work performance. Appropriate and feasible interventions at the health system and individual level need to be introduced urgently to address burnout and depression among HCWs in our setting.

## Acknowledgments

We thank health care workers for accepting to take part in this assessment and for providing important information. We appreciate the research assistants for diligent data collection and express gratitude to the participating health facilities and their management teams for supporting the study.

## Author Contributions

**Conceptualization:** Khumbo Phiri, Corrina Moucheraud, Kathryn Dovel, Risa M. Hoffman, Joep J. van Oosterhout.

**Data curation:** Khumbo Phiri.

**Formal analysis:** John Songo, Hannah Whitehead, Corrina Moucheraud.

**Methodology:** Khumbo Phiri.

**Project administration:** Khumbo Phiri, John Songo, Elijah Chikuse.

**Supervision:** Joep J. van Oosterhout.

**Writing – original draft:** Khumbo Phiri.

**Writing – review & editing:** Hannah Whitehead, Elijah Chikuse, Corrina Moucheraud, Kathryn Dovel, Sam Phiri, Risa M. Hoffman, Joep J. van Oosterhout.

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
