## [Decision Letter · Decision Letter 0]

6 Mar 2023

PGPH-D-22-01998

Burnout and depression among health care workers providing HIV care during the COVID-19 pandemic in Malawi

Dear Dr. Phiri,

Thank you for submitting your manuscript to PLOS Global Public Health. After careful consideration, we feel that it has merit but does not fully meet PLOS Global Public Health’s publication criteria as it currently stands. Therefore, we invite you to submit a revised version of the manuscript that addresses the points raised by the reviewers during the review process.

We look forward to receiving your revised manuscript.

Kind regards,

Rakesh Singh

Academic Editor

Journal Requirements:

1. In the online submission form, you indicated that [Insert text from online submission form here]. All PLOS journals now require all data underlying the findings described in their manuscript to be freely available to other researchers, either 1. In a public repository, 2. Within the manuscript itself, or 3. Uploaded as supplementary information.

While revising the paper, kindly ensure all the journal's requirements.

Reviewers' comments:

Reviewer #1: Thanks for the opportunity to review this paper by Phiri et al. The paper has described the prevalence and factors associated with burnout and depression among health care providers in Malawi. Overall, the paper reads well, however, I have few comments and suggestions as follows:

1. It would be helpful for the authors to detail how the facilities were selected. There is a mention in the abstract that facilities were randomly selected – this is unclear? Was it proportional to HCW population size in the facilities?

2. It is unclear why the authors decided to select a few participants randomly at each of the participating facilities.

3. The sample size estimation should have accounted for a design effect – given the utilisation of multistage sampling approach.

4. It is critical that the analysis to account for the survey design – i.e., the sampling weights, clustering and stratification at the facility level.

5. The analysis of data is unclear. First, it is unclear from the analysis of the data that this study had two main outcomes – the burnout and depression. These should be described separately, especially the multivariable logistic regression analysis. Second, the literature used to determine the factors included in the multivariable model should be cited (included).

6. It is unclear whether the WHO SRQ was validated for use in Malawi.

Some additional minor comments:

1. The authors should consider standardizing the reporting of p values.

2. Include ethics approval reference numbers.

3. Include the implications if the findings (potential interventions , not only screening) in the abstract conclusion section.

Reviewer #2: This is an interesting and well-conducted study. I have a few suggestions to improve the manuscript’s quality and some concerns described below.

INTRODUCTION: Please include a brief summary of the HIV/AIDS epidemic in Malawi - prevalence, number of PLWHA under follow-up & receiving ART etc.

COVID-19 context: This information belongs to your introduction section. Although the authors’ included the absolute number of COVID-19 cases, it would also be important to have the prevalence & mortality rate (if this data is available).

METHODS: Burnout and depression are two mental health disorders that are frequently highly correlated… An analysis of confounding and interaction between the variables should be carried out, as well as a statistical collinearity assessment. I would advise the authors to include a collinearity diagnostic, evaluating the variance inflation factor (VIF) value between SRQ & MBI.

DISCUSSION, 6th paragraph: “Evidence-based treatment for depression, which is feasible in low-resource settings (…).” This statement is controversial. There is a huge treatment gap for mental disorders in low-and middle-income countries (LMIC). LMIC struggle with scarce resources, competing public health priorities, human resource shortages, low investment in mental health services, fragmented service delivery models etc. Those are just a few of the challenges faced by LMIC to properly evaluate the burden of mental disorders and allocate adequate funds/personnel. Please revise your statement.

DISCUSSION: Roughly 30% of study participants screened positive for depression and/or burnout. Undetected and untreated mental disorders are common in LMIC. One key question: Were healthcare workers screened positive for depression and/or burnout referred to additional mental health evaluation and possible treatment? Did they receive prompt mental health counselling after the survey? Did they have access to their screening results? The suicide rate among healthcare professionals during the COVID-19 pandemic was extremely high, especially among those with depression and/or experiencing burnout. What were additional ethical procedures in place for study participants identified as having those mental disorders? Please address this concern.

This study was conducted almost two years ago. I’m curious to know if any knowledge translation and exchange and/or knowledge mobilization strategies were implemented until the submission date, March 2023. For instance, were the study results presented at the selected health facilities? Any brief report was sent to HCW, staff, health clinic directors etc.? During the COVID-19 pandemic, a vast amount of knowledge was created. However, only a fraction of it was effectively mobilized—shared, used and applied—to inform real-life situations. The lack of knowledge transfer or exchange contributes to a gap between research, policy and practice, hindering social innovation and slowing progress in many areas, including global mental health. This is a crucial concern everywhere. But in LMIC, resources are extremely limited. The results of studies such as the one described in this manuscript could inform new practices or (at least) highlight a hidden problem. Please discuss this concern.

---

## [Decision Letter · Decision Letter 1]

7 Jun 2023

PGPH-D-22-01998R1

Burnout and depression among health care workers providing HIV care during the COVID-19 pandemic in Malawi

Dear Dr. Phiri,

Thank you for submitting your manuscript to PLOS Global Public Health. After careful consideration, we feel that it has merit but does not fully meet PLOS Global Public Health’s publication criteria as it currently stands. Therefore, we invite you to submit a revised version of the manuscript that addresses the points raised during the review process.

We look forward to receiving your revised manuscript.

Kind regards,

Rakesh Singh

Academic Editor

Journal Requirements:

a. State what role the funders took in the study. If the funders had no role in your study, please state: “The funders had no role in study design, data collection and analysis, decision to publish, or preparation of the manuscript.”

b. If any authors received a salary from any of your funders, please state which authors and which funders.

Reviewers' comments:

Reviewer #1: Thanks for comprehensive responses to the comments. Note that the p- value should not be reported as 0.00 but rather as <0.01.

Reviewer #3: PGPH-D_22-01998

The research questions required to be formed and justified in the introduction.

The reporting guidelines did not follow the STOBE guidelines, it is highly recommended to use.

Author did the logistic regression; did they check for confounding? Effect modification? Multi-collinearity?

Tools should be reported under subsection in the methods.

Report all the reliability and validity of the tools that were used.

“Descriptive statistics and multivariable

logistic regression models were used to describe the prevalence of screening positive for depression and

of screening positive for burnout.” How can the model describe the prevalence?

Did all assumptions of the regression analysis were met? No information about the assumption of logistic regression I can see.

Before fitting the multivariable regression, the bivariable association requires to check.

In multivariable regression, variable selection requires justification like check of confounding, interaction, and association in bivariate regression along with the support of the literature.

After fitting the model, did they check the model fitness, such as sensitivity analysis, reporting BIC and AIC?

The ROC curve with the AUC should be reported to make sure that the models predicted the outcome.

What is the statistical software used to require reporting?

“All multivariable logistic regression models used standard errors” is it the robust error variance? Require further explanation in the manuscript with reliable citations.

Give information on how the questionnaire was developed and encourage to provide it as a supplementary file.

Authors may find the following articles helpful to correlate their findings with the following articles reported from South Asia:

https://doi.org/10.1371/journal.pgph.0000187

https://doi.org/10.1371/journal.pone.0274965

https://doi.org/10.1016/j.heliyon.2023.e13162

https://doi.org/10.3389/ijph.2022.1604769

https://doi.org/10.1111/inr.12802

https://doi.org/10.1186/s42269-022-00696-1

https://doi.org/10.1186/s43045-021-00103-x

https://doi.org/10.18332/popmed/142558

---

## [Editor Report · Decision Letter 2]

24 Aug 2023

Burnout and depression: A cross sectional study among health care workers providing HIV care during the COVID-19 pandemic in Malawi

PGPH-D-22-01998R2

Dear Ms Phiri,

We are pleased to inform you that your manuscript 'Burnout and depression: A cross sectional study among health care workers providing HIV care during the COVID-19 pandemic in Malawi' has been provisionally accepted for publication in PLOS Global Public Health.

Best regards,

Rakesh Singh

Academic Editor
